# New Approaches and Repurposed Antiviral Drugs for the Treatment of the SARS-CoV-2 Infection

**DOI:** 10.3390/ph14060503

**Published:** 2021-05-25

**Authors:** Luana Vittoria Bauso, Chiara Imbesi, Gasparo Irene, Gabriella Calì, Alessandra Bitto

**Affiliations:** 1Department of Clinical and Experimental Medicine, University of Messina, 98125 Messina, Italy; luanavittoria.bauso@unime.it (L.V.B.); chiara.imbesi@unime.it (C.I.); irene.gasparo@unime.it (G.I.); gabriella.cali@unime.it (G.C.); 2Laboratori Campisi, Corso Vittorio Emanuele 231, 96012 Avola, Italy

**Keywords:** SARS-CoV-2, antivirals, COVID-19, vaccine

## Abstract

Severe acute respiratory syndrome coronavirus 2 (SARS-CoV-2) is the virus that causes coronavirus disease 2019 (COVID-19). The outbreak of this coronavirus was first identified in Wuhan (Hubei, China) in December 2019, and it was declared as pandemic by the World Health Organization (WHO) in March 2020. Today, several vaccines against SARS-CoV-2 have been approved, and some neutralizing monoclonal antibodies are being tested as therapeutic approaches for COVID-19 but, one of the key questions is whether both vaccines and monoclonal antibodies could be effective against infections by new SARS-CoV-2 variants. Nevertheless, there are currently more than 1000 ongoing clinical trials focusing on the use and effectiveness of antiviral drugs as a possible therapeutic treatment. Among the classes of antiviral drugs are included 3CL protein inhibitors, RNA synthesis inhibitors and other small molecule drugs which target the ability of SARS-COV-2 to interact with host cells. Considering the need to find specific treatment to prevent the emergent outbreak, the aim of this review is to explain how some repurposed antiviral drugs, indicated for the treatment of other viral infections, could be potential candidates for the treatment of COVID-19.

## 1. Introduction

Severe acute respiratory syndrome coronavirus 2 (SARS-CoV-2) is the virus that causes coronavirus disease 2019 (COVID-19). The outbreak of this novel coronavirus was first identified in Wuhan (Hubei, China) in December 2019 and it was declared as pandemic by the World Health Organization (WHO) in March 2020.

SARS-COV-2 belongs to the family of Coronaviridae, which are enveloped positive sense single-stranded RNA viruses [1]. The genome size of this viral group ranges between 27 and 34 kilobases, which is much larger than that of most other RNA viruses [2]. When viewed under transmission electron microscopy, coronavirus resembles a crown or the solar corona and its name comes from the Latin word “corona” that just means “crown” or “halo”. Coronaviruses have characteristic club-shaped spike peplomers that surround their surface [3]. The family Coronaviridae is divided into four genera: Alphacoronavirus, Betacoronavirus, Gammacoronavirus and Deltacoronavirus. Alpha- and Betacoronaviruses mainly infect mammals, whereas Gamma- and Deltacoronavirus infect almost exclusively birds [4,5]. To date, seven coronaviruses that can infect humans have been identified, specifically, SARS-CoV, MERS-CoV and SARS-CoV-2 are the most insidious because they infect the lower respiratory tract [6,7]. In severe cases, they cause the acute respiratory distress syndrome (ARDS), that is a potentially fatal condition. SARS-CoV-2 was found to be a novel positive-sense RNA virus, belonging to Betacoronavirus. Like SARS-CoV and MERS-CoV, the genome of SARS-CoV-2 consists of two untranslated regions (UTRs): 5′-cap structure and 3′-poly-A tail [8,9]; in addition, it has distinctive genomic features, including a unique N-terminal fragment within the spike protein. All coronaviruses have their essential genes occurring in the order 5′-S-E-M-N.

The genome of a typical coronavirus contains at least six open reading frames (ORFs) [10,11,12], ORF1a/b is the first and constitutes about two-thirds of the genome, encoding for 16 nonstructural proteins (nsp1–16) [8,9,10,11,12]. A frameshift between ORF1a and ORF1b produces two polypeptides, named pp1a and pp1ab, which are processed by the viral chymotrypsin-like protease (3CLpro) or main protease (Mpro), and by one or two papain-like proteases [13,14]. The sgRNAs are translated to generate the viral structural and accessory proteins. ORFs 10, 11 near the 3′-terminus encode for four main structural proteins spike (S), membrane (M), envelope (E), and nucleocapside (N). In addition, there are other structural and accessory protein such as HE protein, 3a/b protein, and 4a/b protein. The function of these mature proteins is the maintenance of the genome and its replication [13].

Coronavirus membrane displays three or four viral proteins, the glycoprotein (M) is able to span the membrane bilayer three times, it has a short NH2 terminal domain outside and a cytoplasmic domain (COOH Terminus) inside the virus and it is the most abundant structural protein. Conversely the spike S constitutes the peplomers [15]. The ACE receptor expressed on the surface of human cells, represents the target to which the virion Glycoprotein S can attach [16]. Glycoprotein S consists of two subunits: S1 and S2. The S1 subunit through its RBD key domain is responsible for cell tropism and the virus–host range, while the S2 domain through two HR1 and H2R tandem domains allows the fusion of the virus into cell membranes. Following membrane fusion, the viral genome RNA is released into the cytoplasm where it is translated and transcribed [17,18,19]. The RNA-dependent synthesis process comprises two different steps: genome replication with the formation of multiple copies of RNA (gRNA) and transcription of a series of SgmRNA coding for structural and accessory proteins. The protein complex responsible for the continuous and discontinuous synthesis of RNA is encoded by the replicase gene formed by 20 kb. The replicate complex is supposed to consist of 16 viral subunits and a few cellular proteins [20]. In addition to dependent RNA polymerases, it includes helicases, proteases and other enzymes that are commonly absent or rarely found in other RNA viruses such as exoribonuclease specific for the sequence 3′-5′, 20-O-ribose methyltransferase, ADP ribose 10-phosphatase and, in a subset of group 2 coronaviruses, cyclic phosphodiesterase [21,22]. The proteins are assembled at the cell membrane and genomic RNA is incorporated as the mature particle forms by budding from the internal cell membranes [23].

RNA viruses, such as coronavirus, have the capacity of easily changing, therefore giving origin to other varieties. Today there are multiple SARS-CoV-2 variants that circulate in the world. The most important has been identified in the United Kingdom (UK), known as 20I/501Y.V1, VOC 202012/01, or B.1.1.7, and emerged with a large number of mutations. Another variant, found in South Africa and known as 20H/501Y.V2 or B.1.351, appeared independently of B.1.1.7, and finally the most recent emerged in Brazil under the name P.1.

The English variant was first identified in September 2020, and since December some countries, including the United States, have reported several cases of variant B.1.1.7, which shows a mutation in the receptor binding domain (RBD) of the spike protein at position 501, where the amino acid asparagine (N) has been replaced with tyrosine (Y), hence the name of the mutation, known as N501Y. Furthermore, more mutations have appeared in this variant, such as deletion 69/70, which has occurred spontaneously many times and probably leads to a conformational change in the spike protein; the P681H mutation, also spontaneously emerged several times, which is located near the S1/S2 furin cleavage site, a site with high variability in coronaviruses. Recent evidence has reported that this variant is associated with increased transmissibility. Additionally, in January 2021, UK scientists reported evidence suggesting that variant B.1.1.7 may be associated with an increased risk of death.

The South African variant was first identified in Nelson Mandela Bay in early October 2020. B.1.351 has multiple mutations in the Spike protein, including K417N, E484K, N501Y. This variant, unlike the English one, does not contain the deletion 69/70. E484K, the key mutation reported in the literature, confers resistance to neutralizing SARS-CoV-2 antibodies, potentially limiting the therapeutic efficacy of monoclonal antibody treatments [24,25,26,27,28].

The variant P.1 was first identified in Japan in four travelers from Brazil. The Brazilian variant has three mutations located in the Spike protein receptor binding domain: K417T, E484K, and N501Y.

In the literature it is reported that mutations in the P.1 variant could modify its transmissibility and antigenic profile, with repercussions on the ability of antibodies, generated by a previous natural infection or through vaccination, to recognize and neutralize the virus [29].

Currently the reported cases from the WHO globally are nearly 136 million with a total number of deaths of around 3 million (https://www.who.int/emergencies/diseases/novel-coronavirus-2019, accessed on 13 April 2021). The virus may be transmitted by respiratory droplets or by contact with contaminated surfaces (https://www.cdc.gov/coronavirus/2019-ncov/cases-updates/summary.html, accessed on 1 May 2020). The incubation period may vary but is generally reported to be between 1 and 14 days (Lauer et al., 2020). Actually, there are several symptoms reported by positive subjects, among which, the most common are fever (99%), chills, dry cough (59%), production of sputum (27%), fatigue (70%), lethargy, arthralgia, myalgia (35%), anosmia, ageusia, headaches, dyspnea (31%), nausea, vomiting, anorexia (40%), diarrhea. Some subjects may have acute respiratory stress syndrome (ARDS), also in some cases related to past illnesses and death. However, there are test-positive patients who do not have any symptoms. These subjects fall into the category of asymptomatic [30,31].

Today, there are several clinical trials for the use of different vaccines directed against COVID-19, such as inactivated vaccines, nucleic acid-based vaccines and vector vaccines. Currently, several vaccines are being administered to the world population. The first mass vaccination program was launched in December 2020. In fact, on 31 December the WHO organization approved the use of the Pfizer COVID-19 vaccine (BNT162b2), an mRNA vaccine with an efficiency of 95–96%. Later on, another mRNA vaccine, commonly known as Moderna, was approved in January 2020. Finally, the latest to be approved are the vector vaccines known as AstraZeneca and Johnson and Johnson.

One of the key questions today is whether vaccines could preserve against infections from new SARS-CoV-2 variants. Preliminary data support that sera from individuals immunized with SARS-CoV-2 mRNA vaccine neutralize a 501 mutant pseudovirion but less effectively neutralize a 501-487-417 mutant pseudovirion [32]. Furthermore, AstraZeneca was observed to show 83% efficiency for the UK variant but only 22% for the South African variant.

Although vaccines currently in use remain the only effective weapon for the prevention of COVID-19 it appears that they do not allow full coverage for existing variants which is why it is important to focus with the same attention on new therapeutic approaches for the treatment of COVID-19. The main treatment for subjects with a serious infection is oxygen therapy; assisted ventilation is necessary in case of respiratory failure resistant to oxygen therapy and convalescent plasma transfusion [33].

In addition, an emerging and promising therapeutic approach could be the use of monoclonal antibody. Indeed, monoclonal antibodies that target protein S have the potential to prevent SARS-CoV-2 infection and to improve symptoms and limit progression to severe disease in patients with mild to moderate COVID-19. Several monoclonal antibodies against SARS-CoV-2 have been developed and characterized, and four are those approved by the U.S. Food and Drug Administration (FDA): bamlanivimab, etesevimab casirivimab and imdevimab. Recent studies report that patients with the South African variant are resistant to treatment with monoclonal antibodies, for this reason it is necessary to evaluate other therapeutic approaches.

Furthermore, at the beginning of the pandemic some antiparasitic and antimalarial drugs such as ivermectin and chloroquine-hydroxychloroquine, respectively, appeared to be promising for the treatment of COVID-19.

Ivermectin is a macrocyclic lactone, discovered in 1975 and approved by the FDA for parasitic infections. Several studies have reported antiviral activity, in particular against RNA viruses such as Dengue (DENV 1-4), Zika (ZIKV) and the human immunodeficiency virus-1 (HIV-1) because of the inhibition of importin IMP α/β1 heterodimer involved in nuclear import [34]. IMPα/β1 is involved in severe acute respiratory syndrome coronavirus (SARS-CoV) infection [35] and probably it would be a rational aim to consider its efficacy against SARS-CoV-2. A recent in vitro study has demonstrated the efficacy of Ivermectin in reduction of SARS-CoV-2 replication [36]. Currently, there are more than 60 clinical trials to validate the use of ivermectin in COVID-19, although the FDA, today, has not approved it for the treatment or prevention of SARS-CoV-2 infection (https://www.fda.gov/consumers/consumer-updates/why-you-should-not-use-ivermectin-treat-or-prevent-covid-19, accessed on 13 April 2021).

Chloroquine and its derivative hydroxychloroquine are antimalarial drugs that are also used for autoimmune diseases. Despite not being classified as an antiviral drugs, Chloroquine and Hydroxychloroquine have been used for COVID-19 treatment early in the pandemic. In vitro studies demonstrated that chloroquine increases endosomal pH and interferes with glycosylation of cellular receptor of SARS-CoV. A systematic review was performed on the efficacy and safety of chloroquine and chloroquine-related formulations in patients with SARS-CoV-2 pneumonia concluding that there is sufficient preclinical rationale and evidence on chloroquine efficacy and safety due to the use for other indications, but the use of this drug may be supported by expert opinions [37]. In June 2020, the FDA revoked the emergency use authorization (EUA). Based on emerging scientific data, the FDA determined that chloroquine and hydroxychloroquine are unlikely to be effective in treating COVID-19 for the authorized uses in the EUA (https://www.fda.gov/news-events/press-announcements/coronavirus-covid-19-update-fda-revokes-emergency-use-authorization-chloroquine-and, accessed on 13 April 2021).

In contrast, there are other unofficially approved treatments that involve the use of antiviral drugs. Antiviral drugs are those drugs that can interfere both with the life cycle of the virus and on its ability to interact with the host cell. Instances of antiviral drugs are 3CL protein inhibitors (Lopinavir/Ritonavir), RNA synthesis inhibitors (Ribavirin, Remdesivir, Tenofovir Disoproxil Fumarate/TDF and 3TC), neuraminidase inhibitors (Tamiflu and Peramivir) and ACE2 inhibitors [38,39].

Currently there are more than 400 ongoing clinical trials focusing on the use and effectiveness of antiviral drugs as a possible therapeutic treatment (https://www.clinicaltrials.gov/ct2/results?cond=Covid19&term=&cntry=&state=&city=&dist=, accessed on 13 April 2021).

The purpose of this review is to provide a comprehensive overview of some antiviral drugs already tested for the treatment of other viral infections which could be promising in treating COVID-19.

## 2. Monoclonal Antibodies

Bioengineered monoclonal antibodies (mAb) that target Spike protein of SARS-CoV-2 in different sites are currently enrolled in clinical trials.

Casirivimab and imdevimab are recombinant human IgG1 mAbs that bind non-overlapping epitopes of the spike protein of SARS-CoV-2, prevent ACE2 receptor binding and block the virus attachment and entry into human cells. Combination of these two monoclonal antibodies is a “cocktail therapy” produced by Regeneron Pharmaceuticals [40].

In November 2020, the FDA approved the use of Casirivimab and imdevimab for the treatment of adults and pediatric patients suffering from mild to moderate COVID-19, as well as those who are at high risk of progressing to severe COVID-19. The recommended dosage is 1200 mg in a single intravenous infusion. Recent studies have demonstrated that treatment is not recommended in patients who are hospitalized due to COVID-19 or who require oxygen therapy because the drug has limited benefits in patients suffering from severe COVID-19. Issuance of FDA was based on ongoing phase 1/2 double-blind placebo-controlled trial with 799 adults enrolled, in which significant reductions were observed in the viral load (https://www.fda.gov/news-events/press-announcements/coronavirus-covid-19-update-fda-authorizes-monoclonal-antibodies-treatment-covid-19, accessed on 18 March 2021). Other phase 1–2 trials on 275 patients published in December confirm the precedent finding and showed improved results, in fact, the combination of antibodies reduced viral load, in particular in patients whose immune response had not yet been initiated [41]. In addition, today for the use of Casirivimab there are four clinical trials in the recruitment phase (https://clinicaltrials.gov/ct2/results?recrs=&cond=Covid19&term=Casirivimab+&cntry=&state=&city=&dist=, accessed on 18 March 2021).

Bamlanivimab, also known as LY3819253 or LY-CoV555, is a neutralizing IgG1 monoclonal antibody directed against the Spike protein of SARS-CoV-2 developed by Eli Lilly and Company after being identified from a blood sample taken from a COVID-19 patient recovered in North America [42]. In November 2020, the FDA issued EUA for bamlanivimab for the treatment of mild-to-moderate COVID-19 in adult and pediatric patients who are 12 years of age and older weighing at least 40 kg, with a positive COVID-19 test, and with a high risk of progressing to severe COVID-19 and/or to be hospitalized [43]. The EUA is based on data obtained from BLAZE-1, a randomized, double-blind, placebo-controlled, phase 2/3 study on patients with mild to moderate COVID-19, which show that bamlanivimab has direct antiviral activity against SARS-CoV-2, reducing viral load and COVID-related hospitalization. This clinical trial is currently in progress, enrolling a larger cohort of patients [44]. In addition, a randomized, placebo-controlled, double-blind, phase 1 study to evaluate the safety, tolerability, pharmacokinetics and pharmacodynamics in COVID-19 hospitalized patients was completed, while a phase 3 randomized, double-blind, placebo-controlled trial (BLAZE-2), which evaluates the efficacy of bamlanivimab in preventing SARS-CoV-2 infection in residents and staff at nursing and care facilities, is still ongoing. Finally, two phase 2/3 platform studies, ACTIV-2 and ACTIV-3, are evaluating the effect of bamlanivimab in ambulatory and hospitalized with COVID-19, respectively [45]. However, a recent in vitro study showed that bamlanivimab loses its affinity for the receptor binding domain (RBD) of the Spike protein in newly emerging variants from South Africa and Brazil [46]. These data suggest that the use of bamlanivimab could represent a promising pharmacological approach for the treatment of mild-to-moderate COVID-19, although further preclinical and clinical studies are still needed to understand the mechanism of action and the efficacy of bamlanivimab against SARS-CoV-2, especially variants, and to support its use for the treatment of COVID-19 in countries where it has not yet been approved.

Etesevimab, also known as LY-Cov016, is currently being investigated in clinical trials for COVID-19 treatment used with bamlanivimab. Etesevimab is a human and recombinant monoclonal antibody directed against SARS-CoV-2 surface spike protein receptor binding domain. This monoclonal antibody, derived from a patient recovered from COVID-19 in China [42,43,44,45,46,47], can bind with high affinity a SARS-CoV-2 epitope and neutralizes resistant variants with mutations in the mutated epitope recognized by bamlanivimab. A recent study on mild and moderate COVID-19, in non-hospitalized patients in treatment with Etesevimab and Bamlanivimab, has shown significant reduction in SARS-CoV-2 viral load at day 11 unlike Bamlanivimab given in monotherapy [40].

## 3. Antiviral Drugs

### 3.1. Arbidol

Umifenovir (PubChem CID 131410, Figure 1) is a small indole derivative molecule, commonly known by the brand name Arbidol (ARB).

ARB is manufactured as tablets or capsules in Russia and in China, by Moscow-based Masterlek™ (a subsidiary of Pharmstandard Group) and Shijiazhuang No.4 Pharmaceutical™, respectively. In these same countries it is approved for the prevention and treatment of pulmonary diseases caused by influenza and other respiratory viral infections [48]. Several studies have shown that ARB inhibits the fusion of influenzas A and B [49] and other pH-dependent viruses, such as hepatitis B (HBV) [50] and C (HCV) [51] viruses and rhinovirus 14 (HRV-14) [52], within endosomes.

An acidic environment is a fundamental requirement to induce the fusion of these viruses [53], so even a slight increase in pH can repress this process. Thus ARB, with its weak base nature, could raise the endosomal pH and therefore inhibit virus fusion.

However, ARB has also been found to inhibit several pH-independent viruses, such as respiratory syncytial virus (RSV) [47] and parainfluenza virus type 3 [54], suggesting its potential role as broad-spectrum antiviral drug. In fact, several studies in vitro and/or in vivo have pointed out that ARB acts on common phases of the virus lifecycle, and more specifically on cell entry and viral replication of various members of families with a significant clinical impact, such as Orthomyxoviridae, Paramyxoviridae, Picornaviridae, Bunyaviridae, Rhabdoviridae, Reoviridae, Togaviridae, Hepadnaviridae and Flaviviridae [55].

In particular, ARB has been noted to impair the clathrin-mediated endocytosis (CME), an entry mechanism used by several members of viral families including Retroviridae, Adenoviridae, Arenaviridae, Coronaviridae and Togaviridae, by preventing dynamin-2-induced membrane scission, and thereby clathrin-coated vesicle formation [48].

So far, the severe acute respiratory syndrome coronavirus (SARS-CoV) has seemed to be the only one among Coronaviridae that uses CME for its entry into host cells [56,57]. Since SARS-CoV-2 shares a sequence identity of 79.5% with SARS-CoV and binds to the same ACE2 receptor [58], it probably uses the same endocytosis mechanism for entry into host cells. Additionally, a recent structural and molecular dynamics study has shown that Arbidol interacts with a short region of SARS-CoV-2 spike glycoprotein trimerization domain. This result suggests that Arbidol, binding this domain, blocks the trimerization of spike glycoprotein (essential for host membrane fusion) and prevents the viral infection [59].

Based on these lines of evidence, Arbidol seems to have a dual antiviral activity against SARS-CoV-2: by interfering in the binding of the Spike protein with the ACE-2 receptor, it prevents viral binding to host cells; by impairing the mechanism of clathrin-mediated endocytosis, it prevents the fusion and internalization phases of the virus inside the host cells. However, these two mechanisms by which arbidol prevents SARS-CoV-2 infection are still not fully understood and would require further investigation.

Several clinical studies have highlighted the efficacy and tolerability of ARB in the treatment of influenza [60] suggesting a possible use in COVID-19. In addition, a recent retrospective cohort study reports that the clinical response appears more favorable in SARS-CoV-2 patients treated with arbidol combined with lopinavir/ritonavir versus Lopinavir/Ritonavir alone [61]. Subsequently, another clinical study found that Arbidol monotherapy could be more effective than lopinavir/ritonavir for treating SARS-CoV-s2 [62]. Unfortunately, the main limitation of these studies is their small size.

However, all these data suggest that Arbidol could be a potential antiviral intervention in order to immediately prevent the rapid spread of SARS-CoV-2 infection. Currently, two phase IV randomized clinical trials are ongoing to evaluate the efficacy and safety of Arbidol in treating pneumonia in patients with COVID-19, while the mechanism of ARB antiviral activity against SARS-CoV-2, which remains unclear, should be investigated.

### 3.2. Galidesivir

Galidesivir (PubChem CID 10445549, Figure 2), also named BCX4430 or Immucillin-A, is a synthetic adenosine analog designed by BioCryst Pharmaceuticals to inhibit viral RNA-dependent RNA polymerase (RdRp), which plays a critical role in the replication of several RNA viruses, causing a premature RNA chain termination during polymerization [63]. Its efficient uptake and conversion to the active triphosphate nucleotide form by cell kinases allow it to be recognized by the viral RNA polymerase and to be incorporated in place of the corresponding natural nucleotide. Compared to the latter, galidesivir has a carbon instead of nitrogen at position 7 on the base and a nitrogen instead of oxygen at position 1 on the ribose ring. These replacements cause conformational changes which can in turn hinder incorporation of new nucleotides during the chain extension, thereby inhibiting the RNA synthesis [64]. Findings from an in vitro HCV RNA polimerase test suggest that termination happens two nucleotides after incorporation of galidesivir, probably due to stereochemical distortions of the RNA chain in elongation [65]. This mechanism of action makes galidesivir a valid candidate as a broad-spectrum antiviral drug. Galidesivir has shown its antiviral activity against negative-sense RNA viruses involving Filoviridae, Arenaviridae, Bunyaviridae, Orthomyxoviridae, Picornaviridae and Paramyxoviridae families, as well as positive-sense RNA viruses including several members of the Flaviviridae and Coronaviridae families [59]. In vivo studies have shown the high efficacy of galidesivir on several experimental models of viral diseases, including Ebola, Marburg, Rift Valley fever [59], yellow fever [61] and Zika virus infections [66]. In all these studies galidesivir was found to be well tolerated and no evidence of toxicity or adverse reactions was observed. In summary, the broad-spectrum antiviral activity against several virulent RNA viruses, including pathogens such as MERS-CoV and SARS-CoV, combined with the high efficacy and tolerability observed in various preclinical studies, make galidesivir a potential candidate as an antiviral drug for the treatment of the emerging SARS-CoV-2 infection.

A recent molecular docking study has also revealed that the SARS-CoV-2 RdRp model obtained by homology modeling showed a very high sequence identity (about 97%) to the SARS-CoV RdRp and, moreover, galidesivir was able to tightly bind this RdRp model with a binding energy of −7.0 kcal/mol [67].

In April 2020, the recruitment of a randomized, double-blind, placebo-controlled and dose-ranging clinical trial to evaluate the pharmacokinetics, safety and antiviral effects of galidesivir in patients with COVID-19 began and, currently, is in phase I. However, in vitro and in vivo studies to evaluate the antiviral activity of galidesivir against SARS-CoV-2 are still needed.

### 3.3. Nelfinavir

Nelfinavir (PubChem CID 64143, Figure 3) is an antiviral drug belonging to the class of protease inhibitors. The drug is used in the treatment of human immunodeficiency virus (HIV-1), generally in combination with others antiretroviral drugs (e.g., reverse transcriptase inhibitors) [68].

Nelfinavir binds to the active site of the HIV-1 protease and hinders the cleavage of polyproteins precursors encoded by gag and gag-pol genes of HIV (nucleocapsid and core proteins, reverse transcriptase, ribonuclease H, integrase, and protease itself), leading to the formation of new immature viral particles unable to spread the infection to other host cells [69]. Some clinical studies have shown the efficacy of nelfinavir in reducing the HIV viral load below the quantifiable limit (<500 copies/mL) and increasing the mean CD4+ cell count, in addition to its good tolerability. The most common adverse reactions of nelfinavir reported in two large clinical trials were diarrhea, nausea and flatulence [70].

In vitro studies have found that nelfinavir is also able to inhibit herpesvirus replication [71,72] as well as the SARS-CoV replication, in which it performs its inhibitory action during the post-entry phase of replication, but the mechanism is still unclear [73]. These results suggest that nelfinavir may be used as a potential antiviral drug against SARS-associated coronavirus infections. Furthermore, a recent in silico high throughput study reported that the identity sequence of the main protease between the novel SARS-CoV-2 and SARS-CoV is of 96.1% and nelfinavir was identified among the potential drug candidates that tightly bind to SARS-CoV main protease, with binding energy up to −8.6 kcal/mol [74,75]. Finally, a recent in vitro study demonstrated that nelfinavir drastically inhibited SARS CoV-2 Spike glycoprotein-mediated cell fusion, probably binding the Spike trimer structure near the fusogenic domain [76]. Taken together, this evidence suggests the potential use of nelfinavir as an antiviral compound for the treatment of COVID-19, although preclinical and clinical studies evaluating the efficacy of the drug against SARS-CoV-2 are required.

### 3.4. Saquinavir

Saquinavir (SQV, PubChem CID 441243, Figure 4) is an antiviral drug belonging to the protease inhibitor class, it was the first commercially available protease inhibitor for the treatment of HIV infection [77,78]. It is used in combination with other antiretroviral drugs in the so-called HAART (highly active antiretroviral therapy), which includes reverse transcriptase inhibitors and other protease inhibitors. As a protease inhibitor, saquinavir prevents the cleavage of the virus polyprotein precursors (Gag and Gag-Pol) by binding to the HIV protease active site, resulting in the release of immature and noninfectious virions [79]. Saquinavir is mainly metabolized in the liver thanks to the action of cytochrome P450 (CYP), in particular by the specific isoenzyme CYP3A4 [80] but unfortunately, the drug has low bioavailability, therefore is usually administered in combination with ritonavir, which acts as a protease inhibitor but also as a CYP3A4 blocker, thus enhancing the bioavailability of saquinavir [81].

Several clinical trials have also shown that saquinavir exhibits great efficacy and tolerability in the antiviral treatment of HIV-infected patients, with only mild gastrointestinal side effects such as nausea, vomiting, abdominal pain, dyspepsia and diarrhea [82,83]. Previous in vitro and in vivo studies showed that some protease inhibitors typically used in HIV therapy may affect the SARS-CoV main protease [84,85]. On these bases, a recent in silico study also reported that the SARS-CoV protease presents about 96% of homology with the SARS-CoV-2 protease sequence [86]. Taken together, these data suggest that SARS-CoV-2 protease could be a potential target of HIV protease inhibitors.

In addition, more than one molecular docking study found that saquinavir, compared to other compounds, was the drug with the highest binding affinity for SARS-CoV-2 protease, with a binding energy lower than −7.28 kcal/mol [85,86,87]. Nevertheless, several preclinical and clinical studies are still needed to support the use of Saquinavir as an antiretroviral drug for the treatment of the emerging COVID-19.

### 3.5. Favipiravir

Favipiravir (T-705, PubChem CID 492405, Figure 5) is a guanine analogue antiviral drug. It is a pyrazinecarboxamide derivative, a nucleoside precursor, and a prodrug that, once metabolized to its active form (favipiravir-ribofuranosyl-5′-triphosphate), is responsible for the selective inhibition of the viral RNA-dependent RNA polymerase [88]. Other evidence demonstrated that favipiravir induces lethal RNA transversion mutations and stops the viral replication into the growing RNA chain. Favipiravir is active against a large panel of RNA viruses, as observed in several in vitro and in vivo experiments, including Ebola virus, evaluated in a mouse model of infection [89] but also flaviviruses and alphaviruses. In 2014 favipiravir was approved in Japan for the treatment of pandemic flu, but because of its teratogenic effect it cannot substitute current drugs and is prescribed only in situations of emergence. Other in vivo experiments demonstrated that favipiravir has antiviral activity also against West Nile virus, yellow fever virus, foot-and-mouth disease virus, but also arenaviruses, bunyaviruses, and H3N2, H3N2, H5N1 and H1N1 viruses [90]. The catalytic domain of RdRp is conserved in different strains of RNA viruses, in fact favipiravir has a broad spectrum of antiviral activities and probably not only against the influenza virus [91].

The efficacy in stopping viral replication could be used as rational aim for the use of favipiravir in preclinical evaluation for COVID-19. In vitro study on Vero E6 cells infected by COVID-19 has demonstrated the efficacy of favipiravir [31]. Through an open label nonrandomized control study was demonstrated the higher efficacy of favipiravir versus LPV/RTV, but the limit is that it was not a randomized double-blinded placebo controlled clinical trial [92]. Favipiravir is an approved treatment for influenza but currently there is no preclinical support in favor of this antiviral drug as established treatment for SARS-CoV-2. Favipiravir plus interferon alfa was also compared to remdesivir and it received approval from National Medical Products Administration of China thanks to the double effect of inhibition of viral replication and activation of immune system. A prospective, multicenter, open-labeled, randomized clinical trial compared favipiravir to arbidol [93] demonstrating that favipiravir was superior to arbidol in a 7 days observation period, supporting the need of further investigation of the efficacy of favipiravir for the treatment of COVID-19. Currently, there are more than 30 open clinical trials to establish the efficacy and tolerability of this drug. Some of them are in phases 2 and 3, compared with the standard of care, and others in phase 4 compared with hydroxychloroquine.

### 3.6. Remdesivir

Remdesivir (GS-5734, PubChem CID 121304016, Figure 6) is a phosphoramidate 1′-cyano-substituted adenosine nucleotide analogue prodrug [94]. With its antiviral activity, it is widely used for the treatment of viral infections being a potent and selective antiviral drug and is metabolized to active nucleoside triphosphate in several human cell lines.

Remdesivir interferes with the action of viral RNA-dependent RNA polymerase, causing a decrease in viral RNA production [95]. The mechanism of action consists in the insertion into the viral RNA chain, causing an irreversible premature termination. Moreover, remdesivir competes with ATP and it cannot cause an immediate stop, but it will continue to extend the strand with three more other nucleotides. These extra three nucleotides may protect the drug from the activity of viral 3′–5′ exonuclease [96].

Remdesivir was clinically approved as nucleoside analogue and it has been used successfully against RNA viruses, for example hepatitis B virus and human immunodeficiency virus (HIV). In 2016 during the last Ebola virus outbreak, remdesivir was reported to be active in multiple human cell types, in fact it works as an incorporation competitor [97]. Remdesivir was also studied in other severe viral diseases and exhibited an antiviral activity in vitro against Marburg [98], parainfluenza type 3, Nipah, Hendra, Pneumoviridae, and measles and mumps viruses [99]. Remdesivir has been also shown, in in vitro and in vivo models of pulmonary infection, antiviral efficacy against SARS-CoV and MERS-CoV and many other human and zoonotic coronaviruses.

Remdesivir is efficient in chain-terminating, that requires that viral RdRps incorporate nucleotide analogues into the growing RNA strand. The RNA-dependent RNA polymerase, which is named nsp12, conserves the architecture of the polymerase core in different viral strains but a new β-hairpin domain was identified. A comparative analysis model shows how remdesivir binds the polymerase, providing a rationale for the design of new antivirals targeting RdRp [95]. On these bases, remdesivir was proposed as possible therapeutic treatment for COVID-19.

Recent in vitro studies reported the activity against SARS-CoV-2 and demonstrated that remdesivir was highly effective in the control of infection in Vero E6 cells [99] and human liver cancer Huh-7 cells [100]. In a mouse model of SARS-CoV infection, remdesivir induces a decrement of the viral load in the lungs and improved pulmonary function [101]. On February 2020, clinical trials testing efficacy have been officially launched with experimental drugs, and remdesivir showed good safety and pharmacokinetics in I and II phases [102]. It was also tested in multiple trials of different countries, including two randomized phase III trials in China that are expected to be completed in April/May 2020 [103]. In a study published in the New England Journal of Medicine, remdesivir was immediately provided as “Compassionate Use” in patients with severe illness and clinical improvement was observed in 36 of 53 patients [104]. Other studies reported that in case of late initiation of remdesivir therapy, it is however effective in treating SARS-CoV-2 [105]. Currently there are about 40 ongoing clinical trials around the world investigating remdesivir efficacy.

### 3.7. Ribavirin

Ribavirin (PubChem CID 37542, Figure 7) is a synthetic guanosine nucleoside with a broad-spectrum activity against several RNA and DNA viruses. It is a prodrug and, when metabolized, its main direct mechanism of action is to block viral RNA synthesis as polymerase inhibitor. It also interacts with enzymes responsible for ‘capping’ of viral mRNA, but distinct mechanisms of action have been suggested: in fact, an indirect mechanism of ribavirin is to inhibit inosine 5′-monophosphate dehydrogenase thus reducing the level of GTP with a cytostatic effect; it also enhances the T-cell response, although it increases mutation frequency via its incorporation into synthesized genomes [106].

Ribavirin was initially utilized to treat the respiratory syncytial virus in children [107], viral hemorrhagic fever, Lassa fever, measles, and against herpesvirus [108]. Ribavirin has been also effective on other type of infections as the Crimean-Congo hemorrhagic fever, Venezuelan hemorrhagic fever, and Hantavirus infection [109]. After multiple assays in vitro to test the antiviral activity on cell line Vero-RML6 [110] ribavirin was tested on patients with acute respiratory syndrome [111]: in Hong Kong, 138 SARS patients were treated with ribavirin after treatment with oseltamivir [112] while in Canada 126 SARS patients had received it in association with corticosteroids [113].

The activity of ribavirin is not only to interfere with polymerases and although it has a well-established history of usage, now it is indicated to treat hepatitis C in combination with peg-interferon. These peculiar features and the previous use of ribavirin in patients with SARS-nCoV and MERS-nCoV [114] due to the similarities among these infections suggest a possible use for the treatment of COVID-19 [39]. Initially, in China the government recommended the use of ribavirin for COVID-19 but there is no information about its use. Currently about six clinical trials are open to evaluate the efficacy of ribavirin as treatment for SARS-CoV-2. Among these, a prospective open-label randomized controlled trial was completed (phase 2), in which patients were randomly assigned to a 14-day course of either lopinavir/ritonavir, ribavirin and subcutaneous injection of interferon beta-1b (n = 86), versus control group (n = 40) who received lopinavir/ritonavir alone. The results demonstrate that the triple antiviral therapy induces clinical improvement, reduced viral load and alleviation of symptoms in shorter time than control group [115].

### 3.8. Lopinavir/Ritonavir

Lopinavir (PubChem CID 92727, Figure 8A) is a small molecule that contains a hydroxyethylene scaffold and mimics the peptide linkage typically targeted by the HIV-1 protease enzyme, but which cannot be cleaved by the latter, thus preventing the activity of the HIV-1 protease [116]. The HIV-1 protease is a dimeric aspartic protease, a key enzyme that cleaves the Gag polyprotein [117]. Lopinavir is administered in combination with ritonavir, a potent CYP3A inhibitor, to improve lopinavir plasma bioavailability and time exposure of antiviral activity. This combination is approved by the FDA for HIV treatment and is an antiretroviral protease inhibitor that forms an inhibitor-enzyme complex thereby preventing cleavage of the gag-pol polyproteins. Consequently, immature and noninfectious viral particles are produced. Drug targets include nonstructural proteins such as 3-chymotrypsin-like protease, papain-like protease, RNA-dependent RNA polymerase. Molecular docking studies investigate on the possible binding sites among drugs and SARS-CoV-2 structures. A recent analysis through computational methods revealed hydrophobic interaction between SARS-CoV-2 protease and lopinavir with a binding energy of −4.1 kcal/mol.

Lopinavir and ritonavir (PubChem CID 392622, Figure 8B) demonstrated in vitro activity against other novel coronaviruses via inhibition of 3-chymotrypsin-like protease [118,119]. Lopinavir showed an antiviral effect against SARS-CoV-2 virus in Vero E6 cells with the estimated EC50 at 26.63 mΜ [120]. Furthermore, a combination of drugs (lopinavir, oseltamivir and ritonavir) is highly effective against SARS-CoV-2 protease rather than considering each drug separately with binding energy of −8.32 kcal/mol [121]. This in vitro evidence prompted the use of the lopinavir/ritonavir combination in COVID-19; currently there are over 70 clinical trials that investigate its possible efficacy. Cao and colleagues performed a randomized clinical trial using lopinavir/ritonavir in hospitalized adult patients with severe COVID-19 in Wuhan, China, unfortunately no benefit was observed as compared to standard treatment [102]. However, it has been pointed out that recruited patients were late in infection, with severe tissue and lung damage [122] as a matter-of-fact antivirals are most effective when administered in the early phases of an infection [123].

In a study conducted on 47 patients with COVID-19, divided into the test group (lopinavir/ritonavir combined with adjuvant medicine) and the control group (adjuvant medicines alone) it was demonstrated that lopinavir/ritonavir in combination with pneumonia-associated adjuvant drugs, reduced fever, SARS-CoV-2 RNA blood level, and common inflammatory markers faster than in controls [124].

### 3.9. Zanamivir

Zanamivir (PubChem CID 60855, Figure 9) is a small molecule approved for influenza A treatment caused by influenza A virus (strain A/Bangkok/1/1979 H3N2). Zanamivir is a guanido-neuraminic acid that is used to inhibit neuraminidase. Virus neuraminidase catalyzes the removal of terminal sialic acid residues from viral and cellular glycoconjugates [125]. During budding phase, receptor-destroying enzyme neuraminidase cleaves off the terminal sialic acid on the glycosylated HA and virus particles are released. Removing sialic acids from the cell surface, progeny virus can diffuse from cell to cell, thus improving virus spread [126]. In particular, neuraminidase activity improves viral invasion of the upper respiratory tract removing the sialic acid on the mucin of the epithelial cells [127]. Sialidase activity is considerable to enhance virus replication in late endosome/lysosome traffic [125].

An attractive drug target, 3-Chymotrypsin-like protease, has been used to find promising COVID-19 treatment. Protease plays a primary role in replication process and virulence, consequently inhibition of this protein will cause a reduction of infection extent. A recent study shows possible binding sites of drugs and SARS-CoV-2 structures, based on available sequence information, homology modeling and molecular docking studies [86].

SARS-CoV-2 is closely related to the SARS-CoV and this similarity suggests the use of the known protein structures to obtain a model for drug discovery [128]. Hall and colleagues used an in-silico docking model of 3-Chymotrypsin-like protease, responsible for checking several functions of the virus with conserved catalytic domain from the SARS virus [129]. The research has provided the use of Schrodinger Docking Suits^®^ to design the 3-Chymotrypsin-like protease structure. Zanamivir showed high docking score with binding energy of −8.843 kcal/mol. The study reported that Zanamivir has potential to inhibit the SARS-CoV-2 3-Chymotrypsin-like protease. Despite molecular docking studies, a most recent study concerning treatments for coronaviruses states that neuraminidase inhibitors such as zanamivir are not effective and useful for nCoV-2019 [130]. Currently, a clinical trial that employs zanamivir for COVID-19 treatment is not still available.

## 4. Conclusions

The COVID-19 pandemic has caused numerous deaths and it is difficult to predict when it will slow down. Very recently, new cases have been found in China and in other regions where the worst seemed to be gone for good. In the wait for an effective treatment, numerous antiviral pre-existing drugs have been used and, so far, none have been proven really effective, especially in patients with severe alteration of lung function. The main reason is that docking and in silico studies cannot predict the actual drug efficacy that depends on several factors, first of all the viral targets which appear to be unique for SARS-CoV-2. The ongoing trials will provide more answers when completed but, nonetheless, it is imperative to reduce the infection rate with the restriction measures that have been put in place by several governments.

## Figures and Tables

**Figure 1 pharmaceuticals-14-00503-f001:**
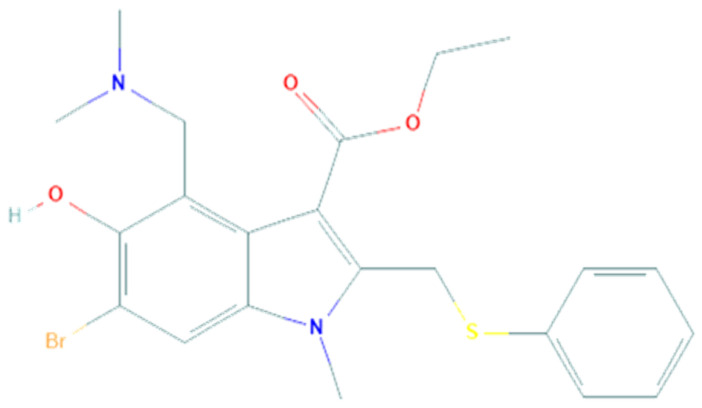
Chemical structure of Arbidol.

**Figure 2 pharmaceuticals-14-00503-f002:**
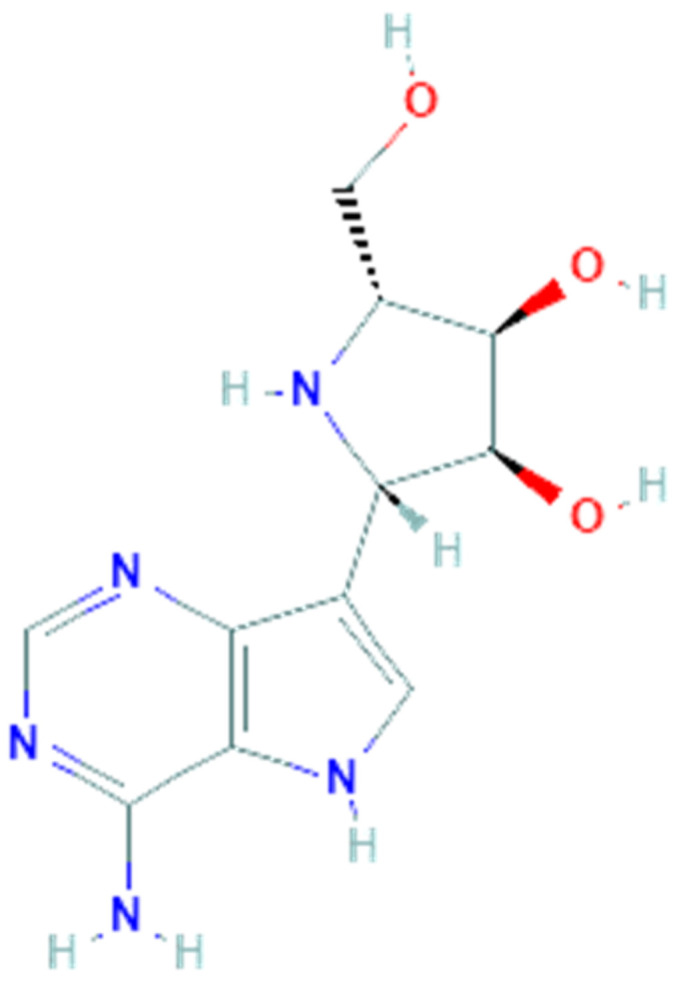
Chemical structure of galidesivir.

**Figure 3 pharmaceuticals-14-00503-f003:**
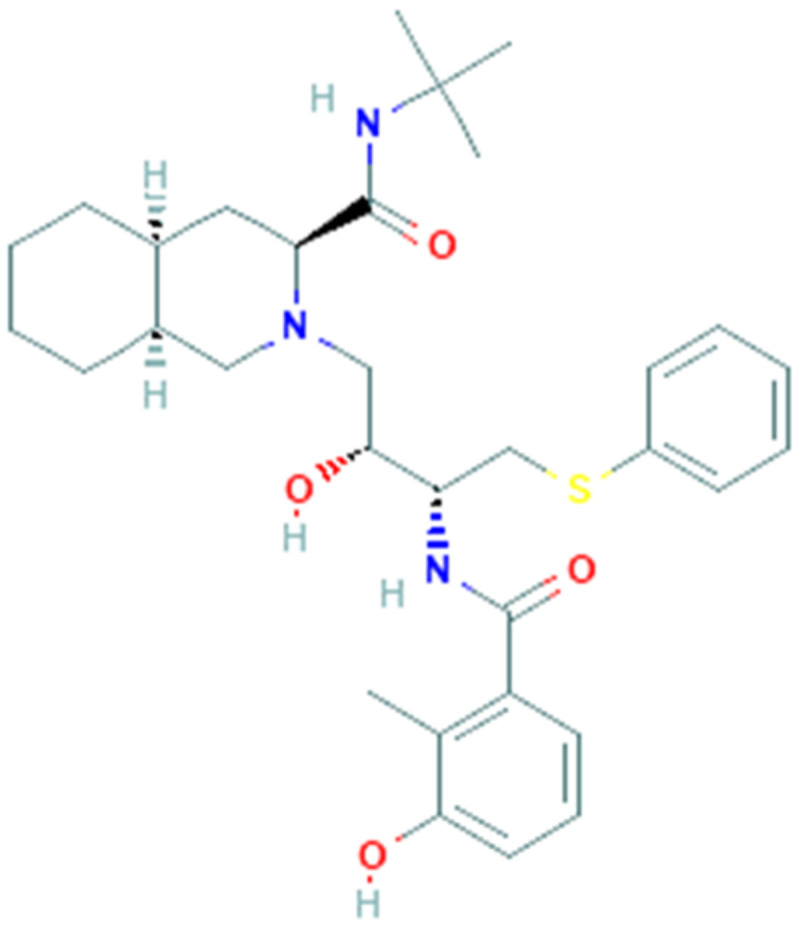
Chemical structure of nelfinavir.

**Figure 4 pharmaceuticals-14-00503-f004:**
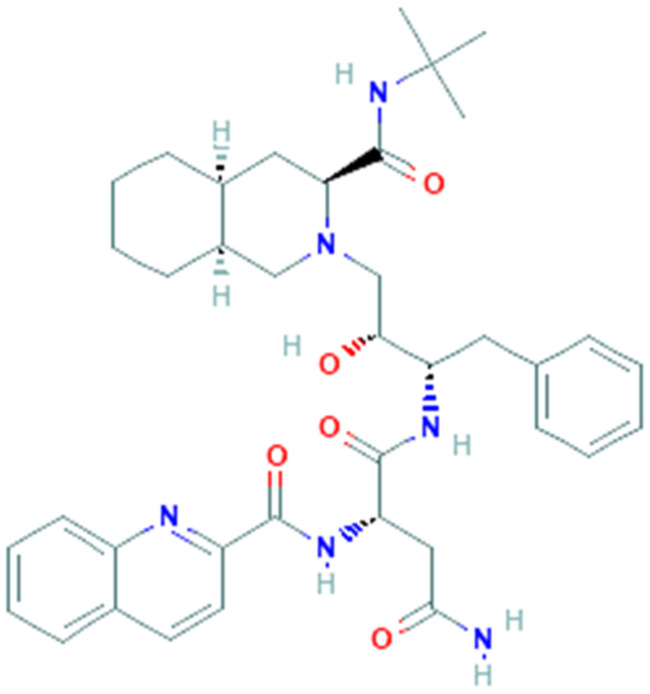
Chemical structure of saquinavir.

**Figure 5 pharmaceuticals-14-00503-f005:**
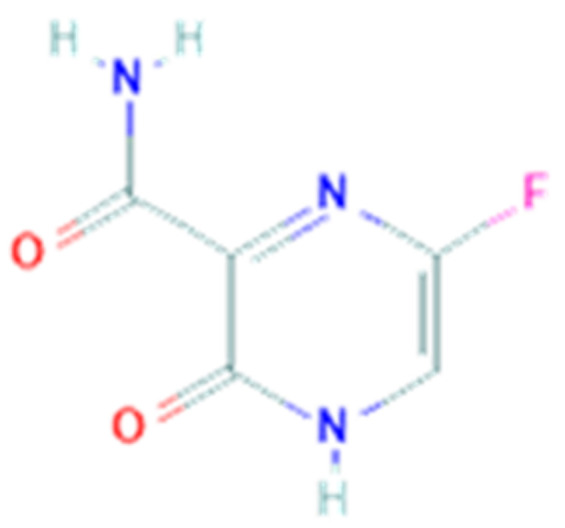
Chemical structure of favipiravir.

**Figure 6 pharmaceuticals-14-00503-f006:**
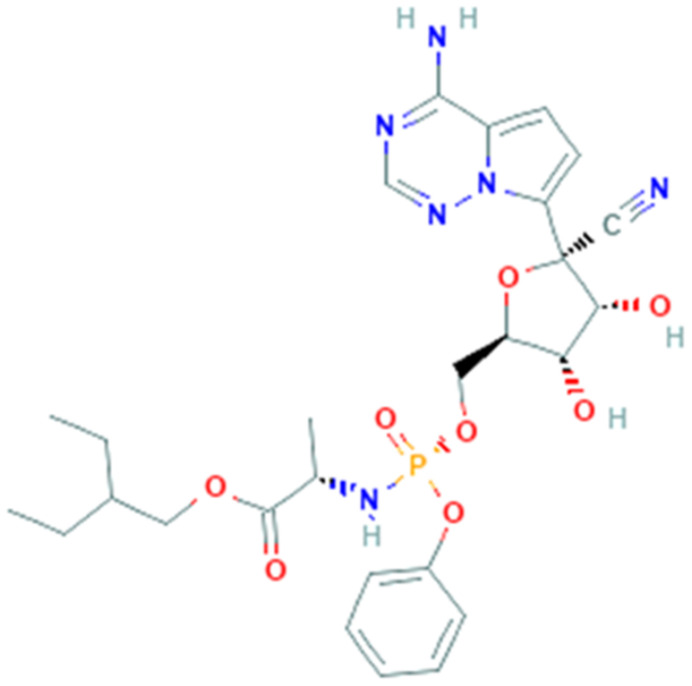
Chemical structure of remdesivir.

**Figure 7 pharmaceuticals-14-00503-f007:**
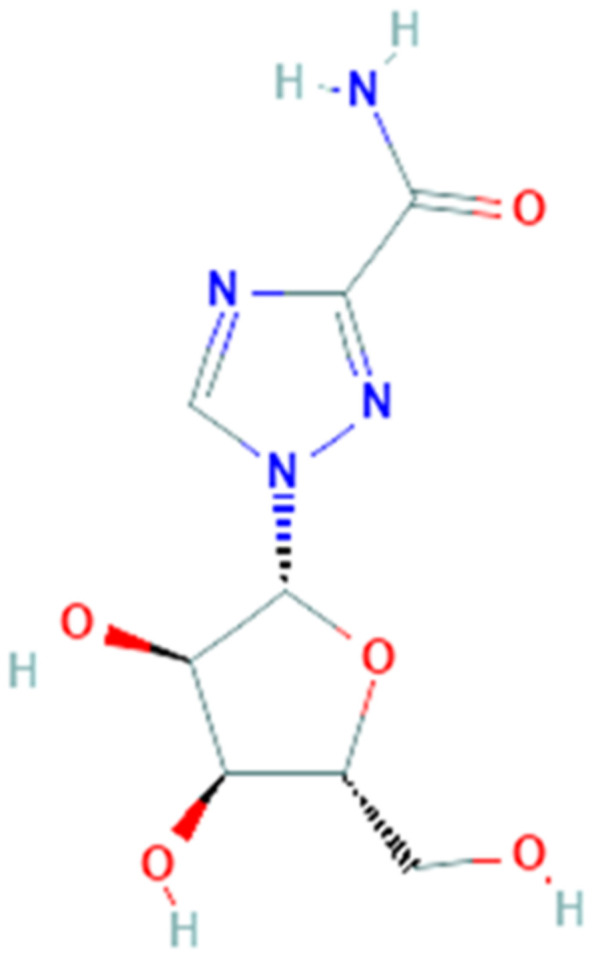
Chemical structure of ribavirin.

**Figure 8 pharmaceuticals-14-00503-f008:**
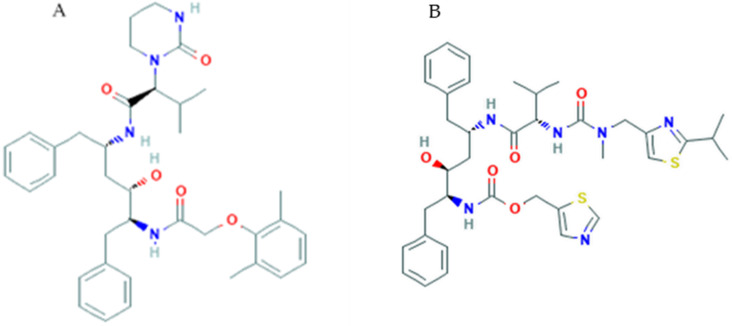
Chemical structure of lopinavir (**A**) and ritonavir (**B**).

**Figure 9 pharmaceuticals-14-00503-f009:**
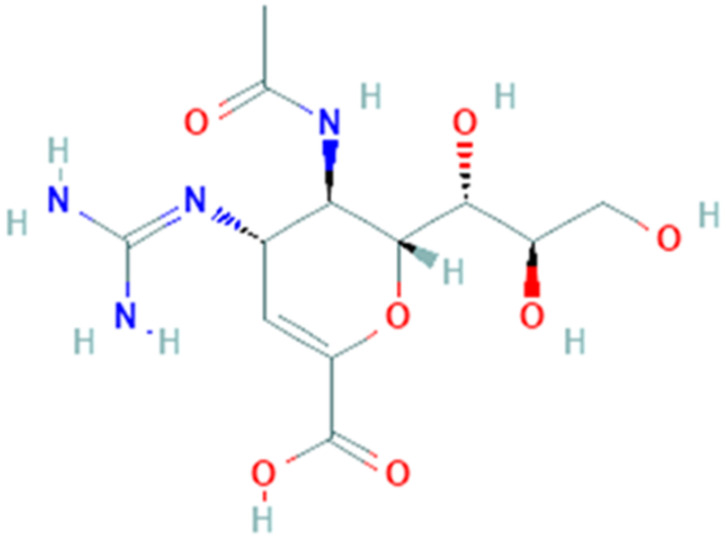
Chemical structure of zanamivir.

## Data Availability

Not applicable.

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
