# Peer review of "New Approaches and Repurposed Antiviral Drugs for the Treatment of the SARS-CoV-2 Infection"

_pharmaceuticals, 2021, doi:10.3390/ph14060503_

Round 1
Reviewer 1 Report
The manuscript is composed in three logically arranged parts: (1) description of the SARS-CoV-2 molecular biology (structure, replication, development – mutations), (2) monoclonal antibodies (mAb) vs COVID-19, (3) antiviral drugs vs COVID-19. The first part deserves a very high rating based on the presentation in a concentrated form of knowledge on SARS-CoV-2, including the latest information on the problem. The second part includes the main attainments of the use in the clinical praxis of the developed prepаrations of monoclonal antibodies targeting the viral spike (S) protein : casirivimab + imdevimab, bamlanivimab and etesevimab. It is known that mAbs were not included in the therapy of SARS-CoV, the first preparation being ready only after the end of a comparatively short SARS epidemics .
Authors of the review underline that all antivirals included in the treatment of COVID-19 have a prehistory and in fact been discovered and described as antivirals against other different viral infections. Authors compose a list of compounds that, on the base of their mechanism of action have the potential to be promising as anti-SARS-CoV-2 drugs. Especially interesting is the description in the first place in Chapter 3 “Antiviral Drugs” of the synthesized and described in Russia arbidol (Leneva et al., 2005). Along the proved as anti-COVID-19 drugs as remdesivir, favipiravir and lopinavir/ritonavir, the authors following their main idea consider in the review as perspective galidesivir, anti HIV nelfinavir and saquinavir, and anti-flu zanamivir. Characterisation of several protease inhibitors (galidesivir, saquinavir, lopinavir/ritonavir, etc.) is illustrated with the respective binding energy values.
Remarks:
-In the review MS along with the proved anti-SARS-CoV-2 substances, ivermectin (Cally et al., 2020) is omitted.
-Chloroquine and hydroxychloroquine are not included in the review, evidently because of their anti-COVID-19 activity being not proved, nevertheless that European Medicines Agency issued guidance that these substances are only to be used in clinical trials or emergency use programs (Corteliani et al., 2020).
Author Response
Specific comments:
- In the review MS along with the proved anti-SARS-CoV-2 substances, ivermectin (Cally et al., 2020) is omitted.
Chloroquine and hydroxychloroquine are not included in the review, evidently because of their anti-COVID-19 activity being not proved, nevertheless that European Medicines Agency issued guidance that these substances are only to be used in clinical trials or emergency use programs (Corteliani et al., 2020).
As suggested, ivermectin, chloroquine and hydroxychloroquine were included with the references recommended by the Reviewer, considering that at the beginning of pandemic these drugs were be evaluated as promising for emergency use. However, despite studies and ongoing clinical trials reported in our manuscript, we pointed out that FDA did not approve ivermectin and revoked chloroquine and hydroxychloroquine for COVD-19 treatment. For this reason, our work mainly focuses on the re-proposed antiviral drugs which, based on the rationale we have explained, could be promising for the treatment of COVID-19.
Reviewer 2 Report
There comes another review on the treatment of SARS-CoV-2. As the new coronavirus swept across the world, antiviral research has drawn attention and taken a fast-paced development like never before. As the speed up to thousands of original research papers per month, a high-quality, up-to-date review paper could really help to navigate the current situation for researchers or the general public. This review has provided extensive knowledge for the COVID-19 and the general development of antiviral research, as has been done by many other similar reviews. But unfortunately, some of the information is rather dated and somehow biased. The authors stated the current vaccine has low efficiency at 22% towards the new variants to emphasize the urgency for antiviral therapy but ignore the fact that a vaccine with 22% efficiency still has 80% protection from the severe COVID cases. In contrast, many antivirals listed have already been proved with no effect against CoV-19 (such as Ribavirin, due to exonuclease activity. Also, Ribavirin is mistaken as protease inhibitors in Line-159); Arbidol is presented with the antiviral mechanism from preventing viral entry through the endocytosis, while the dual-entry mechanism for CoV-19 is well-established, which leaves the argument defective. Scientific review, unlike an opinion news article, should provide information from all perspectives. I suggest the paper be reorganized and revised before consider publishing. Chemical structure instead of chemical formula should be included; Many other editorial errors should also be checked and proofread.
Author Response
Specific comments:
- The authors stated the current vaccine has low efficiency at 22% towards the new variants to emphasize the urgency for antiviral therapy but ignore the fact that a vaccine with 22% efficiency still has 80% protection from the severe COVID cases.
In agreement with the Reviewer’s comment, although the AstraZeneca vaccine has, as reported in our manuscript, only 22% efficacy against the South African variant, the vaccine's coverage against severe cases of COVID-19 should not be underestimated. For this reason, we have highlighted how, despite the low efficacy against some variants, the current vaccines represent the only weapon capable, today, of preventing the viral transmission and the onset of COVID-19.
- Ribavirin is mistaken as protease inhibitors in Line-159.
As noted by the Reviewer, there is a typo in Line-159, which was corrected by placing ribavirin in the right category of antiviral drugs (RNA synthesis inhibitors).
- Arbidol is presented with the antiviral mechanism from preventing viral entry through the endocytosis, while the dual-entry mechanism for CoV-19 is well-established, which leaves the argument defective.
As pointed out by the Reviewer, the entry mechanism of SARS-CoV-2 has been clarified. However, it is not yet fully understood in which specific life cycle stage of SARS-CoV-2 Arbidol acts. Therefore, in our manuscript we have pointed out how the two modes of action of Arbidol described, on the basis of the rationale explained in our work, are both plausible and would require further investigations to be well established.
- Chemical structure instead of chemical formula should be included.
In agreement with the Reviewer's suggestion, the chemical formula has been replaced with the corresponding chemical structure of the antiviral drugs described in our manuscript. The chemical structures have been retrieved from the PubChem database and reported in the manuscript as figures. In addition, the Compound ID number (CID) of each antiviral drug has been added.
- Many other editorial errors should also be checked and proofread.
We apologize to the Editor and Reviewers for any editorial errors found in the text, and we guarantee that these have been corrected.
We hope that now everything is in order and we look greatly forward your positive reply. We wish to express our deepest appreciation for your skillful comments to our work.
With best regards,
Professor Alessandra Bitto